# Scoping review of hearing loss attributed to congenital syphilis

**Aleena Amjad Hafeeez[1], Karina Cavalcanti Bezerra[2], Zaharadeen Jimoh [3], Francesca B. Seal [4], Joan L. Robinson [5]\*, Nahla A. Gomaa[6]**

1 Department of Pediatrics, Queen's University, Kingston, Ontario, Canada, 2 Research Investigator, University of Alberta, Edmonton, Alberta, Canada, 3 Department of Nursing, University of Alberta, Edmonton, Alberta, Canada, 4 Department of Anesthesiology, University of Alberta, Edmonton, Alberta, Canada, 5 Department of Pediatrics, University of Alberta, Edmonton, Alberta, Canada, 6 Division of Otolaryngology-Head & Neck Surgery, University of Alberta, Edmonton, Alberta, Canada

* jr3@ualberta.ca

**Data Availability Statement:** All data are in the manuscript and/or supporting information files.

**Funding:** The author(s) received no specific funding for this work.

## Abstract

### Background

There are no narrative or systematic reviews of hearing loss in patients with congenital syphilis.

### Objectives

The aim of this study was to perform a scoping review to determine what is known about the incidence, characteristics, prognosis, and therapy of hearing loss in children or adults with presumed congenital syphilis.

### Eligibility criteria

PROSPERO, OVID Medline, OVID EMBASE, Cochrane Library (CDSR and Central), Proquest Dissertations and Theses Global, and SCOPUS were searched from inception to March 31, 2023. Articles were included if patients with hearing loss were screened for CS, ii) patients with CS were screened for hearing loss, iii) they were case reports or case series that describe the characteristics of hearing loss, or iv) an intervention for hearing loss attributed to CS was studied.

### Sources of evidence

Thirty-six articles met the inclusion criteria.

### Results

Five studies reported an incidence of CS in 0.3% to 8% of children with hearing loss, but all had a high risk of bias. Seven reported that 0 to 19% of children with CS had hearing loss, but the only one with a control group showed comparable rates in cases and controls. There were 18 case reports/ case series (one of which also reported screening children with hearing loss for CS), reporting that the onset of hearing loss was usually first recognized during adolescence or adulthood. The 7 intervention studies were all uncontrolled and published in

**Competing interests:** The authors have declared that no competing interests exist.

1983 or earlier and reported variable results following treatment with penicillin, prednisone, and/or ACTH.

## Conclusions

The current literature is not informative with regard to the incidence, characteristics, prognosis, and therapy of hearing loss in children or adults with presumed congenital syphilis.

## Introduction

Syphilis is a sexually transmitted infection caused by the bacterium *Treponema pallidum*. If not recognized and treated early in pregnancy, fetal transmission commonly occurs [1]. According to the international Joint Committee on Infant Hearing, congenital syphilis (CS) is a risk indicator for hearing loss [2,3]. The Centers for Disease Control and Prevention state: "Otosyphilis is caused by an infection of the cochleovestibular system with *T. pallidum* and typically presents with sensorineural hearing loss, tinnitus, or vertigo. Hearing loss can be unilateral or bilateral, have a sudden onset, and progress rapidly." (Neurosyphilis, Ocular Syphilis, and Otosyphilis (cdc.gov)). Almost all cases of CS are treated with penicillin which is not known to be ototoxic.

For decades, congenital syphilis had almost disappeared in Canada and the United States due to low rates of syphilis in the community and universal prenatal screening.. The number of cases of confirmed early congenital syphilis born to women aged 15–39 years in Canada rose from 17 cases in 2018 to 117 in 2022 [4]. Trends in the United States (US) mirror this with an increase from 1325 congenital syphilis cases in 2018 to 3755 in 2022 [5].

The recent resurgence has increased interest in the clinical manifestations and complications of congenital syphilis. There are no published data summarizing the incidence or characteristics of hearing loss due to congenital syphilis. Despite the larger number of cases now occurring in Canada and the US, there are no evidence-based guidelines on screening or management of hearing loss in children with congenital syphilis. We therefore performed a scoping review. Our specific questions were:

- How often is hearing loss due to congenital syphilis?

- What is the incidence of hearing loss in children with congenital syphilis?

- When hearing loss occurs from congenital syphilis, what is the usual age of onset? Is it unilateral or bilateral? How severe is it? How rapidly does it progress?

- Is there evidence for any interventions for treatment of hearing loss attributed to congenital syphilis?

This will inform the studies that need to be done to determine the incidence and age of onset of hearing loss from CS, the severity of hearing loss, and interventions that warrant further study.

## Methods

The methodology was based on the Preferred Reporting Items for a Systematic Review and Meta-analysis Extension for Scoping Reviews: The PRISMA-ScR statement [6] (See attached S1 Checklist). A search was executed by a health librarian on the following databases:

PROSPERO, OVID Medline, OVID EMBASE, Cochrane Library (CDSR and Central), Proquest Dissertations and Theses Global, and SCOPUS using controlled vocabulary (e.g.: MeSH, Emtree, etc.) and selecting key words representing the concepts "congenital syphilis" or "hearing loss" (S1 Appendix). Databases were searched from inception to October 17, 2021, with an updated search to March 31, 2023.

Articles were included if they described persons of any age with hearing loss that the authors of the article attributed to congenital syphilis. To delineate the burden and incidence of hearing loss from congenital syphilis, we included any studies that i) screened children with hearing loss for evidence of congenital syphilis or ii) screened children with congenital syphilis for hearing loss. We also included randomized controlled trials (RCTs), cross-sectional studies, case series, and case reports that described the characteristics of hearing loss, the long-term outcomes of hearing loss, or the results of any interventions for hearing loss. We excluded autopsy reports, animal studies, studies focusing solely on acquired syphilis and those published in a language other than English, French, or Portuguese.

Articles published in English were screened by two reviewers independently [AH, KC], and conflicts were resolved by a senior author [JR, NG]. Articles published in French had a single reviewer [FS]. There were no articles published in Portuguese. Because of the small number of recent articles, preprints were included. The protocol has not been published.

Studies were divided into four types: i) those that screened patients with hearing loss for congenital syphilis, ii) those that screened patients with congenital syphilis for hearing loss, iii) case reports or case series that describe the characteristics of hearing loss in patients with congenital syphilis, and iv) studies that describe an intervention for hearing loss attributed to congenital syphilis. Data were collected and managed using Research Electronic Data Capture (REDCap) tools [7] hosted at the University of Alberta with the extracted data determined by the study type. Data were entered by a single investigator. The JBI critical appraisal tool was used as appropriate to assess all included studies [8–11] (S2 Appendix). The critical appraisal and bias risk assessment was completed by a single reviewer [NG], and all studies were rated as high, unclear or low risk of bias.

## Results

The search yielded 1983 records of which 832 were duplicates. Screening led to 159 records for full-text review of which 36 met inclusion criteria (Fig 1). The figure outlines the reasons for exclusion of other records.

### Screening of patients with hearing loss for congenital syphilis

There were 5 studies where patients with hearing loss were screened for CS. They were published from 1900 to 1990 and all had a high risk of bias (Tables 1 and 2). The incidence of CS ranged from 0.3% to 8% in children attending schools for the hearing impaired and was 2% in children seen at a clinic for the hearing impaired.

### Screening of patients with CS for hearing loss

There were 7 studies of which 4 were published from 2016 to 2022 (Table 3). The risk of bias was high for 1, unclear for 3, and low for 3. Hearing loss was reported in 0 to 19% of children with probable or proven CS. One study from the modern era showed an incidence of 6% (22/342) (12). However, a small recent study reported no hearing loss for 7 infants treated in utero, a 5% incidence for 37 treated at birth, and a 6% incidence in 49 controls [23].

**2021 Hafeez/Gomaa Congenital Syphillis and Hearing Loss**

3rd August 2023

covidence

**Fig 1. PRISMA flow diagram.**

## Case series and case reports of hearing loss attributed to CS

There were 10 case series (one of which was also included in Table 2) (Table 4A) and 8 case reports (Table 4B) of which all but 6 were published prior to 1980. The risk of bias was high for 5 articles, unclear for 3 and low for 10. In these reports, hearing loss was often first noted in adolescence or adulthood with the youngest being 5 years old at diagnosis. Many cases also had interstitial keratitis. Follow-up was too variable to allow determination of the expected

**Table 1. Critical appraisal of studies of hearing loss attributed to congenital syphilis.**

| Type of study | Author (Year) Country | Risk of bias | Comments |
|---|---|---|---|
| Children with HL screened for CS | Gururaj [12] (1900) India | High | • Quota Sampling<br>• Descriptive statistics<br>• Method of diagnosis of CS or HL not described |
| | Yearsley [13] (1910) UK | High | • Only the demographics and phenotypes of patients in case reports described<br>• Method of diagnosis of CS or HL not described<br>• Inclusion or exclusion criteria not clear<br>• Longitudinal reporting and follow up not defined |
| | Wright [14] (1968) UK | High | • Quota Sampling<br>• Descriptive statistics - no correlation between CS and HL proven<br>• Method of diagnosis of CS or HL not described |
| | Kameswaran [15] (1976) India | High | • Judgmental non-probability sample<br>• Method of diagnosis of HL subjective [audiometry]<br>• Descriptive statistics - no correlation between CS and HL proven |
| | Ganga [16] (1990) India | High | • Judgmental sampling<br>• Descriptive statistics - no correlation between CS and HL proven<br>• Main objective was to look at the psychological impact rather than the etiology of HL |
| Children with CS were screened for HL | Keidel [17] (1923) US | Moderate | • Adequate sample size<br>• Setting described<br>• Descriptive study with no data analysis |
| | Ittkin [18] (1953) US and Canada | Low | • Adequate sample size<br>• Study subjects and setting described<br>• Sufficient data with statistical analysis<br>• Identification of the condition valid |
| | Gleich [19] (1994) US | Moderate | • A quota sample used<br>• ABR used for auditory screening<br>• Descriptive study<br>• Correlation could not be proved |
| | Simms [20] (2016) UK | Moderate | • A quota sample was used<br>• Definition of CS precise but no clear definition of HL |
| | Lim [21] (2021) South Korea | High | • A quota sample used<br>• No clear diagnostic criteria or tools for HL<br>• Though they applied statistics, not clear what they were trying to prove |
| | Besen [22] (2022) Brazil | Low | • Adequate sample size<br>• Study subjects and setting described<br>• Sufficient data and analysis<br>• Fails to describe how CS diagnosed |
| | Aguiar [23] (2022) Brazil | Low | • Adequate sample size<br>• Study subjects and setting described<br>• ABR used for auditory screening. Note that equipment was changed during study but no statistical significance.<br>• Sufficient data and statistical analysis<br>• Identification of condition valid |
| Case series | Yearsley [13] (1910) UK | High | • Only the demographics and phenotypes of patients in case reports were described.<br>• Method of diagnosis of CS or HL not described<br>• Inclusion or exclusion criteria unclear<br>• Longitudinal reporting and follow up not defined |
| | Fraser [24] (1916) UK | High | • Patients demographics and phenotypes were described.<br>• No diagnostic tests used to confirm HL |
| | Perlman [25] (1952) US | Moderate | • Demographics, and audio-vestibular loss described. |
| | Morton [26] (1955) UK | Moderate | • Demographics, CS diagnosis were described.<br>• Audio vestibular symptoms depicted more than diagnostics. |
| | Morton [27] (1957) UK | Moderate | • Brief demographics and symptomatology<br>• Audio-vestibular diagnosis not reported<br>• Method of CS diagnosis not described |
| | Baron [28] (1967) France | High | • No patient's demographics, patient's history, clinical condition, or invention described<br>• only the diagnostic tests or assessment methods in addition to intervention and post intervention clinical conditions<br>• Adverse effects and objectives not clear. |

(*Continued*)

**Table 1.** (Continued)

| Type of study | Author (Year) Country | Risk of bias | Comments |
|---|---|---|---|
| | Dawkins [29] (1968) UK | Low | • Demographics and diagnostics well described |
| | Patterson [30] (1968) US | Low | • Demographics and diagnostics well described |
| | Kerr [31] (1970) Ireland | Low | • Brief demographics<br>• Methods of CS diagnosis described |
| | Indesteege [32] (1989) Belgium | Low | • Brief demographics<br>• Audio vestibular diagnosis of Leutic Meniere's disease explained<br>• Method of CS diagnosis explained |
| Case reports | Hastings [33] (1915) US | Low | • Demographic described well<br>• Audio-vestibular finding explained<br>• Longitudinal results of treatment assessed |
| | Murphy [34] (1958) US | Low | • Demographic described well<br>• Audio-vestibular finding explained<br>• Longitudinal results of treatment assessed |
| | Karmody [35] (1966) US | Low | • Demographic described well<br>• Audio-vestibular finding explained<br>• Histopathology added when autopsy performed |
| | Hughes [36] (1981) Germany | Low | • Demographic described well<br>• Audio-vestibular finding explained<br>• Longitudinal results documented |
| | Khetarpal [37] (2011) US | High | • Demographic described<br>• More emphasis on other manifestation than on HL |
| | Pessoa [38] (2011) Brazil | High | • Demographics described<br>• Symptoms and signs of CS explained including HL<br>• Not all hearing assessments included |
| | Singhal [39] (2011) India | Low | • Method of diagnosis of CS described<br>• Also had a cleft palate that could have contributed to HL. |
| | Kivekas [40] (2014) US | Low | • Demographics described. This is a case report of radiological findings in the temporal bones because of congenital syphilis.<br>• Symptoms and signs were explained as well as interventions in addition to radiologic findings. |
| Studies of an intervention | Hahn [41] (1961) US | Moderate | • Small number/ cohort<br>• Confounding factors were not accounted for<br>• It was mentioned that longitudinal results are difficult to obtain |
| | Kerr [42] (1973) Ireland | Moderate | • Patients lost to follow up not accounted for<br>• Longitudinal follow up unclear |
| | Smyth [43] (1976) Ireland | Low | • Exclusion criteria mentioned. |
| | Zoller [44] (1979) US | Moderate | • Detailed description of demographics and clinical picture<br>• Audiometric results before and after treatment provided<br>• No group comparison |
| | Wilson [45] (1981) US | Moderate | • Confounding factors not accounted for<br>• Longitudinal results unclear |
| | Dobbin [46] (1983) US | Moderate | • Stats purely descriptive<br>• No mention of confounding factors |
| | Kerr [47] (1983) Ireland | Low | • Adverse effects mentioned<br>• Conclusion articulated |

CS–congenital syphilis; HL–hearing loss; UK–United Kingdom; US–United States.

**Table 2. Studies where multiple children with hearing loss were screened for congenital syphilis.**

| Author (Year) Country | Description of participants screened for CS | Number of children with CS |
|---|---|---|
| Gururaj [12] (1900) India | 300 children with HL at a school for children with visual and hearing impairment | 15 (5%) |
| Yearsley [13] (1910) UK | 225 children with acquired HL in "deaf centres" in London with acquired HL (did not test 229 with congenital HL or 46 with unclear timing of onset of HL) | 17 (8%) |
| Wright [14] (1968) UK | 1157 patients with SNHL at the Royal National Throat, Nose, and Ear Hospital 1963 to 1966 | 20 (2%) |
| Kameswaran [15] (1976) India | 367 children at schools for the hearing impaired | 1 (0.3%) |
| Ganga [16] (1990) India | 100 children at schools for the hearing impaired | 1 (1%) |

CS–congenital syphilis; HL–hearing loss; SNHL–sensorineural hearing loss.

**Table 3. Studies where children with congenital syphilis were screened for hearing loss.**

| Author (Year) Country | Population screened | Participants with HL | Hearing loss pattern |
|---|---|---|---|
| Keidel [17] (1923) US | 230 with CS seen at Syphilis Clinic at John Hopkins Hospital 1917–1923. | 23 (10%) | Late SNHL was found in 22 patients plus one had deafness due to gummatous condition in nasopharynx. Onset occurred from 7 to 27 years of age. Four had sudden HL (over the course of a week.). Seven had bilateral HL. |
| Ittkin [18] (1953) US and Canada | 59 with late CS | 11 (19%) | Two had sudden onset HL, 3 had gradual onset, with no comment on the other 6. Three had excessive high tone HL and 8 had subnormal HL (defined as narrowed range to voice, forks and audiometric tests with impaired bone conduction |
| Gleich [19] (1994) US | 75 neonates with positive FTA-ABS or MHA-TP and Apgar > 9 at 5 minutes of life (provide no data on how many had CS versus passive antibodies– 41 had CSF obtained and none had evidence for CNS syphilis) | 0 | NR |
| Simms [20] (2016) UK | 17 patients born 2010–2015 with possible or proven CS | 1 (6%) | NR |
| Lim [21] (2021) South Korea | 250 neonates treated for possible or proven CS | 34 (14%) including 6 of 14 (43%) with neurosyphilis | NR 14% had hearing impairment according to The International Classification of Diseases-10 codes |
| Besen [22] (2022) Brazil | 21,434 newborns evaluated in a Brazilian reference hearing health service of the UNHS Program. | 22 (6%) | NR |
| Aguiar [23] (2022) Brazil | 93 newborns divided into three groups. Group 1: Prenatal treatment for syphilis (n = 7) Group 2: Treatment for syphilis after birth (n = 37) Group 3: Control group with no syphilis (= 49) | Group 1: 0/7 Group 2: 2/37 (5%) Group 3: 3/49 (6%) | NR |

CSF–cerebrospinal fluid; CNS–central nervous system; CS–congenital syphilis; HL–hearing loss; NR–not reported.
SNHL–sensorineural hearing loss.

**Table 4.** A - Case series of hearing loss attributed to congenital syphilis. B - Case reports of hearing loss attributed to congenital syphilis.

| Author (Year) Country | Demographics | Hearing loss assessment | Interventions |
|---|---|---|---|
| Yearsley [13] (1910) UK | 17 cases with onset at 6 to 14 years (unknown for 2 cases). All had interstitial keratitis (preceded deafness in at least 10 cases). | Bone conduction, otoscopy, traditional ear exam -used subjective unconfirmed behavioral measures. | NR |
| Fraser [24] (1916) UK | 33 cases between the ages of 7 and 38 with HL recognition at 3,6,7 (n = 2), 8 (n = 4), 9 (n = 2), 10 (n = 2), 11 (n = 3),12,14, 15 (n = 2), 16, 17, 18, 19 (n = 2), 20, 24, 25, 27 (n = 2), 21(n = 2), 30, and 33 years. HL sudden (N = 8), gradual (N = 14) and unknown (N = 11). All but one had interstitial keratitis. | Bone conduction, otoscopy, vestibular irritation test. | One case received potassium iodide application and pilocarpine injection with no improvement |
| Perlman [25] (1952) US | 11 cases with recognition of HL loss at 9, 11, 12, 17 (N = 2), 18, 19, 23, 33, and 35 (N = 2) years. All had interstitial keratitis. | All but one had bilateral HL. Cold caloric test abnormal in 10 of 11 and fistula test positive in 2 of 9 where results reported. | Bismuth, arsenic +/- penicillin given to 5 with no improvement–all 5 then given ACTH and 3 had improvement |
| Morton [26] (1955) UK | 4 cases with onset at approximately ages 15, 34, 35 and 53 years | Traditional audiometry inconsistently mentioned | All received beta-pyridylcarbinol, and 2 of 4 received penicillin– 3 reported subjective improvement |
| Morton [27] (1957) UK | 4 cases with onset at approximately 18, 24 and 35 years (unknown for one patient) | Unclear | -Penicillin treatment in 3 of 4 cases. Three received prednisone 10 mg three times daily for 3 days followed by 5 mg twice daily for 3 weeks. One reported subjective improvement. |
| Baron [28] (1967) France | 3 children aged 10, 11 and 16 years with maternal diagnosis made after diagnosis in child | Traditional audiometry: moderate bilateral HL (N = 1), minimal bilateral hearing loss (N = 1), mild HL on right and minimal on left | Case 1: Penicillin 80 million U and mercury cyanide. Deterioration of hearing bilaterally. Case 2: Multiple course antibiotics. Improvement of hearing bilaterally. Case 3: Removal of vegetations, radiotherapy of skull, tubal massages, and antibiotics. Deterioration of hearing on the left and improvement on the right |
| Dawkins [29] (1968) UK | 5 cases with onset of HL at age 13, 20, 42, 37 and 55 years | Traditional audiometry: Moderate HL in right ear and moderate-severe in left (N = 1), minimal HL in right ear and mild HL in left (N = 1), mild HL bilaterally (N = 1), profound HL in right and moderate HL in left (n = 1) and mild HL in right and severe in left (N = 1) | All treated with prednisone of which 3 responded: Case 1 right ear improved. Case 3 had a slight response but only to pure tone audiogram and case 5 had moderate improvement that was not maintained |
| Patterson [30] (1968) US | 4 cases with onset of HL at age 26,32 29, and 39 years) | Limited results reported | All received prednisone and reported some improvement in symptoms with 2 having documented improved speech discrimination but 3 prescribed maintenance prednisone as improvement transient |
| Kerr [31] (1970) Ireland | 3 cases aged 46 to 55 years with onset of HL at ages 26, 47 and 53 years | Pure tone threshold and speech discrimination: Left ear HL followed by right ear HL with vertigo (N = 1), fluctuating HL with rotational vertigo (N = 1) and bilateral HL (N = 1) | All cases: high dose penicillin for four weeks, prednisone for ten days with ten-day taper. All reported to have improved speech discrimination in at least one ear and vertigo improved in both cases |
| Indesteege [32] (1989) Belgium | 5 cases aged 50 to 61 years at time of study (not clear when HL started) | Audiometry, tympanometry and stapedial reflex. Found to have symmetrical HL with Meniere's disease and tinnitus (N = 3), mixed, bilateral perceptive, symmetric, and fluctuant HL with Meniere's disease and tinnitus with vertigo (N = 1) and bilateral perceptive and fluctuant HL with Meniere's disease, tinnitus and vertigo | 2 cases treated with corticosteroids but discontinued due to side-effects |

*(Continued)*

**Table 4.** (Continued)

| Author (Year) Country | Patient demographics | Diagnosis of congenital syphilis | Hearing loss assessment | Interventions |
|---|---|---|---|---|
| Hastings [33] (1915) US | 20-year-old female with Hutchinson's triad | Maternal diagnosis prenatally, but asymptomatic for 19 years. | Traditional audiometry | Mercurial topical treatment |
| Murphy [34] (1958) US | 20-year-old female with HL diagnosed at age 9 years | Age 9 years | Traditional audiometry with mild mixed HL, more marked in the right ear. | Multiple treatments with penicillin and prednisone |
| Karmody [35] (1966) US | 44-year-old male with bilateral HL diagnosed at age 18 | Maternal diagnosis after birth | Traditional audiometry with profound HL in right ear and severe HL in left ear | Penicillin at age 32 and then at age 44 with no improvement |
| Hughes [36] (1981) Germany | 39-year-old male with bilateral HL fluctuating on the right | Unreported | Traditional audiometry | Penicillin 500000 U, probenecid, and prednisolone |
| Khetarpal [37] (2011) US | 5-year-old with bilateral hearing loss with early and late features of CS | 4 months–ruled out neurosyphilis | No reported hearing assessment | Penicillin at 10 months of age |
| Pessoa [38] (2011) Brazil | 7-year-old female with Hutchison's triad | 2 years | NR | Penicillin–unknown dose |
| Singhal [39] (2011) India | 14-year-old female with Hutchison's triad | 14 years | NR | Benzathine penicillin 2.4 million units intramuscularly |
| Kivekas [40] (2014) US | 75-year-old female with bilateral hearing loss since age 10 | Positive syphilis diagnosis after chronic interstitial keratitis at unknown age–in 1950s | No reported hearing assessment. 76% bisyllabic work scores with right ear cochlear implant. | None |

CSF–cerebrospinal fluid; CNS -central nervous system; HL–hearing loss; NR–not reported; UK–United Kingdom: US–United States.

HL–hearing loss; NR–not reported.

rate of progression of hearing loss. A wide variety of therapies are reported with small numbers of patients and inconsistent results that were often subjective.

## Studies with interventions for hearing loss

The 7 studies included a range of 6 to 39 patients with the most recent one being from 1983 (Table 5). All were observational. Most commonly patients were prescribed penicillin with addition of prednisone followed by ACTH if response was poor or transient. Outcomes were often subjective and inconsistent. Risk of bias was unclear for 5 studies and low for 2 studies.

## Discussion

The scoping review shows that studies of hearing loss due to congenital syphilis are limited and low quality. All but one study reported as a pre-print [23] are observational studies and only 15 of 36 studies (42%) were at low risk of bias. One cannot determine the incidence or characteristics of hearing loss from congenital syphilis or the efficacy of interventions from this review. It seems unlikely that a systematic review would find further studies that could answer these questions.

**Table 5. Studies with an intervention for hearing loss that was attributed to congenital syphilis with minimal details on individual patients.**

| Author (Year) Country | Total participants | Intervention | HL characteristics prior to intervention | HL characteristics following intervention |
|---|---|---|---|---|
| Hahn [41] (1961) US | Group A: Patients with syphilitic neural deafness (N = 19) Group B: Patients with non-syphilitic neural deafness (N = 20) | Prednisone 30 mg daily in divided doses for one week, then 20 mg daily, then 2.5 mg taper per day each month if progressing well. Six patients received penicillin at the start of prednisone therapy. | Group A: On average, lower discrimination scores prior to treatment. Group B: Cases described often had sudden HL with decreases in discrimination scores noted. | Group A: Definite improvement in discrimination score occurred for eight patients. Two patients had slight increases in discrimination scores. The other 9 patients saw no significant improvements. Group B: Very slight improvements in 3/20 cases, but overall, no improvements noted |
| Kerr [42] (1973) Ireland | 12 | All got ampicillin, at least 3 also got prednisone, 4 of whom also got ACTH. | Case 1: Fluctuations in pure tone hearing and speech discrimination for 2–3 years prior to treatment. Case 2 and 3: Profound deafness for greater than 6 months with speech discrimination score of 0. Case 4: High speech discrimination score, but low pure tone hearing score with sudden drop in speech discrimination prior to treatment. | All had improved speech discrimination. |
| Smyth [43] (1976) Ireland | 19 | Group A (N = 9): Ampicillin for 4 weeks. Group B (N = 13 –includes 3 who failed from Group A): ampicillin plus prednisone 10 mg TID for 10 days (if no improvement additional 10 days of prednisone at same dose). Group C (N = 8, all of whom failed in Group B): ampicillin, prednisone and ACTH 40 to 120 units per week (duration of treatment not specified) | Group A: 6 ears initially had low speech discrimination scores; 9 ears had satisfactory hearing. Group B: 11 ears had profoundly HL and 12 had significant HL. Group C: 6 ears with profound and 8 ears with significant HL. Overall: 2 patients had severe tinnitus and 5 patients had vertigo. | Group A: 6 ears which initially had low speech discrimination scores improved by 20% in speech discrimination score–was only maintained in 2 ears. 9 ears had satisfactory hearing and maintained their hearing. Group B: All initially improved but 7 ears relapsed. Group C: All improved. All but one had serviceable hearing with follow-up of maximum 6.5 years. |
| Zoller [44] (1979) US | 12 | Penicillin G plus 80 mg prednisone every other day | Mean SRT in left ear was 54.5 dB and 51.0 dB on the right. Mean speech discrimination score was 47.2% on the left and 61.6% on the right | 3 patients had sustained improvement in HL at 1 year, 1 had transient improvement, and 5 had no improvement in HL. |
| Wilson [45] (1981) US | 7 | 3 months of weekly benzathine penicillin G and 80 mg of prednisone orally every other day up to one month. | All had vertigo | Vertigo improved in 4 and unchanged in 3. |
| Dobbin [46] (1983) US | 13 | Penicillin G IV daily for 2 weeks followed by benzathine penicillin every other week for 10 weeks. Prednisone was co-administered starting at 80 mg every other morning for 1 month and tapered based on patient response. | Unknown | 9 of 26 ears showed a response to treatment. 2 ears showed improvements to pure tone threshold while 8 ears showed improvements (one ear demonstrated improvements in both). 5 ears demonstrated transitory improvements discrimination Only 4 ears had any lasting response in speech discrimination or pure tone threshold. |
| Kerr [47] (1983) Ireland | 17 | Group A (N = 1): Ampicillin for 4 weeks Group B (N = 7): Ampicillin and prednisone 10 mg for 10 days with 10-day taper. Group C (N = 6): Ampicillin, prednisone, and ACTH (40 units to 120 units) for prolonged course Group D (N = 3)–deceased Followed for mean 10.5 years | Group A: Fluctuating HL. Group B: Mean speech discrimination score 62%. 4 ears had profound, 6 had fluctuating HL and 4 were stable. Group C: Mean speech discrimination score 46%. 3 ears had profound and 8 had fluctuating HL. | Group A: Fluctuating HL, speech discrimination score remained normal. Group B: No response for those with profound HL. 4 of 6 ears with fluctuating HL remained fluctuating with no changes to speech discrimination score. 2 remained stable and 1 ear had an improvement in speech discrimination score from 28% to 56%. For the 4 stable ears prior to treatment, 2 remained stable, 1 developed profound and 1 developed fluctuating HL post-treatment. Group C: No response for those with profound HL. Out of those with fluctuating HL, 1 remained stable and 1 progressed to profound HL. The other 6 continue to have fluctuating hearing loss. |

ACTH - Adrenocorticotropic hormone; HL–Hearing loss; PTA–pure tone average; SRT–speech recognition threshold.

As expected, there were major variations in the study methodologies employed to diagnose hearing loss. In the early 1900s, investigators used basic tuning fork tests and subjective behavioral responses [24]. Studies performed after the year 2000, used full diagnostic tests or Auditory Brainstem Responses (ABR) for neonates [21].

A small percentage of children attending schools for the hearing impaired had evidence of congenital syphilis. However, these data are of limited value without a control group from the same jurisdiction. The percentage of hearing loss that is due to congenital syphilis no doubt varies considerably by country and over time.

It is perhaps unexpected that almost all case reports and case series describe recognition of hearing loss only in adolescence or adulthood. It is possible that hearing loss started years prior but was not recognized, particularly, if the hearing loss was slowly progressive. The major problem with all these reports is that they do not exclude the possibility that the patient had acquired syphilis or had another etiology for their hearing loss.

Clearly, there is paucity of up-to-date literature regarding this important health problem. The majority of articles were published before 1980. The recent surge in congenital syphilis cases in Canada and the United States may lead to further studies. Recent results from neonatal hearing screening programs in low- or middle-income countries where the incidence of congenital syphilis never waned are informative. Besen reported screening 21,434 newborns in Brazil 2017 through 2019 and reported a prevalence of test failure in the Universal Neonatal Hearing Screening Program (UNHS) of 1.6% (95% CI: 1.4; 1.8). This study used Otoacoustic Emission and ABR to identify both cochlear and retrocochlear damage. They report that 1.7% (95% CI: 1.5; 1.8) had congenital syphilis but do not report how many with congenital syphilis had hearing loss [22]. In a follow-up report of 34,801 infants screened 2017 through 2021, they report that neonates with congenital syphilis were 2.38 times as likely to fail in the UNHS as those without congenital syphilis [48]. However, another small study from Brazil reported as a pre-print examined failed hearing screens at 2 months of life did not find an association between congenital syphilis and failed hearing screens [23].

It is not clear whether there is a treatment for hearing loss due to congenital syphilis. Antibiotics were presumably always given at the time of diagnosis of hearing loss if the patient had not previously been adequately treated. There are no convincing reports that this alone resulted in sustained improved hearing. Uncontrolled studies that included corticosteroids with or without ACTH reported variable response and improvement in hearing was often subjective.

The main limitation of this scoping review is the lack of high-quality studies.

## Conclusion

Our scoping review outlines a general map of the trend of publications across the decades and shows that the incidence of hearing loss due to congenital syphilis is completely unknown. It is not clear whether the stage of maternal syphilis or the age at which infants are treated changes outcomes. The literature does not inform us as to whether treatment in-utero prevents development of hearing loss. Until there are high quality long-term observational studies, it is difficult to know what hearing screening to recommend for children with congenital syphilis. Hearing loss attributed to congenital syphilis is often first recognized in adolescence or adulthood. Therefore, there is a need to increase awareness that people of all ages with unexplained hearing loss of sudden or gradual onset should be screened for syphilis. Other than treatment of the congenital syphilis, no other treatments can be recommended until there are RCTs or cohort studies with valid control groups.

## Supporting information

**S1 Checklist. Preferred Reporting Items for Systematic reviews and Meta-Analyses extension for Scoping Reviews (PRISMA-ScR) checklist.**
(DOCX)

**S1 Appendix. Systematic review search strategy.**
(DOCX)

**S2 Appendix. Joanna Briggs Institute (JBI) critical appraisal checklist.**
(DOCX)

## Author Contributions

**Conceptualization:** Joan L. Robinson, Nahla A. Gomaa.

**Data curation:** Aleena Amjad Hafeeez, Karina Cavalcanti Bezerra, Zaharadeen Jimoh, Francesca B. Seal, Nahla A. Gomaa.

**Formal analysis:** Karina Cavalcanti Bezerra, Joan L. Robinson.

**Project administration:** Nahla A. Gomaa.

**Supervision:** Joan L. Robinson, Nahla A. Gomaa.

**Writing – original draft:** Aleena Amjad Hafeeez, Joan L. Robinson, Nahla A. Gomaa.

**Writing – review & editing:** Aleena Amjad Hafeeez, Karina Cavalcanti Bezerra, Zaharadeen Jimoh, Francesca B. Seal, Joan L. Robinson, Nahla A. Gomaa.

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
