## [Decision Letter · Decision Letter 0]

22 Nov 2023

PONE-D-23-32597

Scoping review of hearing loss attributed to congenital syphilis.

PLOS ONE

Dear Dr. Robinson,

Thank you for submitting your manuscript to PLOS ONE. After careful consideration, we have decided that your manuscript does not meet our criteria for publication and must therefore be rejected.

I am sorry that we cannot be more positive on this occasion, but hope that you appreciate the reasons for this decision.

Kind regards,

Sangamanatha Ankmnal Veeranna, Ph.D.

Academic Editor

PLOS ONE

Additional Editor Comments:

The authors have put a lot of effort into this article. But the manuscript is not conveying the important message of this scoping review. A lot of key information is missing in the introduction, results, and the discussion section.

What is the purpose of this scoping review?

What are the authors trying to convey from this scoping review? Do the authors think that there should be a systematic review conducted based on a scoping review?

How does congenital syphilis cause hearing loss? Hearing loss is the sequelae? or medicine given to treat syphilis causes hearing loss?

What is the current gap in the knowledge base on congenital syphilis?

What is the available evidence in individuals with congenital syphilis?

I did not see the overview or the map of evidence on congenital syphilis.

How was the hearing research conducted in the past and currently?

Reviewers' comments:

Reviewer's Responses to Questions

**Comments to the Author**

1. Is the manuscript technically sound, and do the data support the conclusions?

Reviewer #1: Yes

2. Has the statistical analysis been performed appropriately and rigorously? 

Reviewer #1: N/A

3. Have the authors made all data underlying the findings in their manuscript fully available?

Reviewer #1: Yes

4. Is the manuscript presented in an intelligible fashion and written in standard English?

Reviewer #1: Yes

5. Review Comments to the Author

Reviewer #1: Title of the Manuscript:

"Scoping Review of hearing loss attributed to congenital syphilis."

Overall Evaluation:

The paper offers valuable insights into hearing loss attributed to congenital syphilis. The authors put a lot of effort into finding the relevant literature and conducting the scoping review. Overall, I find the work satisfactory. However, I find changes are required to the text. There were a few spelling and grammar mistakes. I recommend improving the organization of the content in the paper.

Major Contributions:

Relevance of Topic: The topic is of high importance and is aptly chosen.

Comprehensiveness: The review covers a wide array of studies and provides a comprehensive overview.

Originality: The topic that the authors selected is a novel research area. There is adequate information; however, I suggest improvements to further identify gaps and mention potential areas for future research.

Areas for improvement:

A document with suggestions for improvements is attached. Please refer to the document. The document also contains the paper with corrections and questions marked in red. Please refer.

Summary and Recommendation:

Overall, this manuscript is a commendable effort in synthesizing the current knowledge on hearing loss attributed to congenital syphilis. It provides an insightful overview, though some areas require refinement for clarity and depth. With revisions, particularly in the introduction, methodology, and discussion sections, this manuscript will be a valuable addition to the literature. I recommend acceptance of the manuscript after revisions.

6. PLOS authors have the option to publish the peer review history of their article (what does this mean?). If published, this will include your full peer review and any attached files.

Reviewer #1: No

- - - - -

---

## [Author Response · Author response to Decision Letter 0]

19 Dec 2023

Thank-you for agreeing to review a revision of our manuscript. We have addressed the comments of the reviewers as follows:

A. Editor’s Comments:

The authors have put a lot of effort into this article. But the manuscript is not conveying the important message of this scoping review. A lot of key information is missing in the introduction, results, and the discussion section.

1. What is the purpose of this scoping review?

Response: The submitted version of the manuscript stated in the introduction: “There are no published data on the incidence or characteristics of hearing loss from CS. We therefore performed a scoping review to better characterize this entity. This will inform the studies that need to be done to determine the incidence and age of onset of hearing loss from CS, the severity of hearing loss, and interventions that warrant further study”. 

To make our objectives clearer, we added the following to this paragraph:

Our specific questions were:

• How often is hearing loss due to congenital syphilis?

• What is the incidence of hearing loss in children with congenital syphilis?

• When hearing loss occurs from congenital syphilis, what is the usual age of onset? Is it unilateral or bilateral? How severe is it? How rapidly does it progress?

• Is there evidence for any interventions for treatment of hearing loss attributed to congenital syphilis? 

 We also added a sentence to make it clearer that there are no guidelines for ongoing management of hearing loss with CS. 

2. What are the authors trying to convey from this scoping review? Do the authors think that there should be a systematic review conducted based on a scoping review?

Response: We agree that we should have made it clearer that the scoping review showed a great need for higher quality studies. We have added the following sentence to the first paragraph in the discussion: “One cannot determine the incidence or characteristics of hearing loss from CS or the efficacy of interventions from this review. It seems unlikely that a systematic review would yield answers to any of these questions.”

3. How does congenital syphilis cause hearing loss? Hearing loss is the sequelae? or medicine given to treat syphilis causes hearing loss?

Response: We agree that we could have explained this better and added to the first paragraph: “The Centers for Disease Control and Prevention state: “Otosyphilis is caused by an infection of the cochleovestibular system with T. pallidum and typically presents with sensorineural hearing loss, tinnitus, or vertigo. Hearing loss can be unilateral or bilateral, have a sudden onset, and progress rapidly.” Penicillin is the only treatment used and it is not known to be an ototoxic drug. We now state “Almost all cases of CS are treated with penicillin which is not known to be ototoxic.”

4. What is the current gap in the knowledge base on congenital syphilis?

Response: As noted above, the introduction of the submitted version of the manuscript stated: “There are no published data on the incidence or characteristics of hearing loss from CS.”

5. What is the available evidence in individuals with congenital syphilis?

Response: It is not clear what the editor is asking. Our scoping review summarizes the evidence on hearing loss. Did they want us to talk about all aspects of congenital syphilis?

6. I did not see the overview or the map of evidence on congenital syphilis.

Response: We used the Cochrane methods for our scoping review (See Scoping reviews: what they are and how you can do them | Cochrane Training ). I do not think that an overview or a map of evidence are a mandatory part of a scoping review. A review article on evidence mapping (Miake-Lye IM et al. What is an evidence map? A systematic review of published evidence maps and their definitions, methods, and products. Syst Rev. 2016 Feb 10:5:28. doi: 10.1186/s13643-016-0204-x) states: “… scoping reviews include “a descriptive narrative summary of the results” whereas evidence maps identify evidence gaps, and both use a tabular format to depict a summary of literature characteristics “. Given the lack of evidence we uncovered, it would be very difficult to make a useful evidence map. 

7. How was the hearing research conducted in the past and currently?

Response: We are sorry but we could not determine what the editor is asking. We tell the reader that all but one study were observational studies so there is not a specific methodology. 

B. Reviewer’s Comments:

Overall Evaluation:

The paper offers valuable insights into hearing loss attributed to congenital syphilis. The authors put a lot of effort into finding the relevant literature and conducting the scoping review. Overall, I find the work satisfactory. However, I find changes are required to the text. There were a few spelling and grammar mistakes. I recommend improving the organization of the content in the paper.

Major Contributions:

Relevance of Topic: The topic is of high importance and is aptly chosen.

Comprehensiveness: The review covers a wide array of studies and provides a comprehensive overview.

Originality: The topic that the authors selected is a novel research area. There is adequate information; however, I suggest improvements to further identify gaps and mention potential areas for future research.

Areas for improvement:

A document with suggestions for improvements is attached. Please refer to the document. The document also contains the paper with corrections and questions marked in red. Please refer.

Summary and Recommendation:

Overall, this manuscript is a commendable effort in synthesizing the current knowledge on hearing loss attributed to congenital syphilis. It provides an insightful overview, though some areas require refinement for clarity and depth. With revisions, particularly in the introduction, methodology, and discussion sections, this manuscript will be a valuable addition to the literature. I recommend acceptance of the manuscript after revisions.

1. An explicit statement of research questions and objectives is not provided. I recommend rethinking and clearly rewriting the text. Please use bullet points and proper formatting. It helps the reader to understand clearly.

Response: Thanks for pointing out that our objectives were not as clear as we thought that they were. We added the following to the introduction: 

Our specific questions were:

• How often is hearing loss due to congenital syphilis?

• What is the incidence of hearing loss in children with congenital syphilis?

• When hearing loss occurs from congenital syphilis, what is the usual age of onset? Is it unilateral or bilateral? How severe is it? How rapidly does it progress?

• Is there evidence for any interventions for treatment of hearing loss attributed to congenital syphilis? 

2. State the process for selecting sources of evidence (i.e., screening and eligibility) included in the scoping review. Reviewer says: The selection process is clear. However, I recommend re-think and clearly describe in the text. Use bullet points and proper formatting. Formatting issues were identified.

Response: I am sorry but it was not clear what the reviewer wanted us to change. It is not common to use bullet points when describing the sources that were searched for a systematic or scoping review. 

3. For each source of evidence, present characteristics for which data were charted and provide the citations. Reviewer states: Charts are included. I recommend moving the charts below to relevant sections in the text. Otherwise, it was challenging to scroll down and go back to the text to relate what the authors are trying to describe.

Response: I apologize for this error. I clearly did not read the author instructions carefully. The tables have been moved. 

4. Discussion - Found complex sentence structures. Look into the text again and think of clear explanation. 

Response: We tried to make the text clearer throughout the discussion. 

5. The reviewer asked that we tell the reader the number of articles included in the “Design” section of the abstract. 

Response: It is customary not to divulge results until the “Results” section of the abstract. 

6. The reviewer suggests adding a reference to the sentence “Syphilis is a sexually transmitted infection caused by the bacterium Treponema pallidum.” 

Response: It seems unlikely that the reader would require a reference to believe that this statement is correct. We can add one if required. 

7. We were asked to double-check whether the statistics that we quote in the introduction from previous studies are correct.

Response: We have no way of verifying this but have quoted the studies accurately. 

8. Methods – We were asked where the keywords are. 

Response: As far as we can tell, PLOS ONE does not request keywords. We can add some if required. 

9. Articles published in English were screened by two reviewers independently [AH, KC], and conflicts were resolved by a senior author [JR, NG]. Articles published in French had a single reviewer [FS]. There were no Portuguese articles. Because of the small number of recent articles, preprints were included. The protocol has not been published.´The reviewer stated that perhaps we did not need to include this information and asked whether the protocol was in Prospero.

Response: We can omit this information if requested. The protocol is not in Prospero as unfortunately they only publish protocls for systematic reviews – not scoping reviews. 

10. Figure 1 is far away from the text.

Response: Instructions to authors request that figures be submitted as a separate file. 

11. “ One study from the modern era showed an incidence of 6% (22/342) (Besen et al., 2022).” Reviewer states: “not required or explain in words”, I think referring to 22/342.

Response; We think that it is helpful for the reader to not have to look at the table to determine the sample size for this statistic, but can omit that if requested. 

12. Multiple minor changes were made in the text as requested.

---

## [Decision Letter · Decision Letter 1]

16 Feb 2024

PONE-D-23-32597R1Scoping review of hearing loss attributed to congenital syphilis.PLOS ONE

Dear Dr. Robinson,

Thank you for submitting your manuscript to PLOS ONE. After careful consideration, we feel that it has merit but does not fully meet PLOS ONE’s publication criteria as it currently stands. Therefore, we invite you to submit a revised version of the manuscript that addresses the points raised during the review process.

We look forward to receiving your revised manuscript.

Kind regards,

Bolajoko O. Olusanya, MBBS, FMCPaed, FRCPCH, PhD

Academic Editor

PLOS ONE

Journal Requirements:

When submitting your revision, we need you to 
address these

additional requirements.

1. Please ensure that your manuscript meets PLOS ONE's style

requirements, including those for file naming. The PLOS ONE style templates can

be found at

and

2. We note that your Data Availability Statement is

currently as follows: This is a systematic

review so all data can be derived from the included manuscripts.

Please confirm at this time whether or not your submission

contains all raw data required to replicate the results of your study. Authors

must share the “minimal data set” for their submission. PLOS defines the

minimal data set to consist of the data required to replicate all study

findings reported in the article, as well as related metadata and methods

(https://journals.plos.org/plosone/s/data-availability#loc-minimal-data-set-definition).

- The values behind the means, standard deviations and other

measures reported;

Authors do not need to submit their entire data set if only

a portion of the data was used in the reported study.

If your submission does not contain these data, please

either upload them as Supporting Information files or deposit them to a stable,

public repository and provide us with the relevant URLs, DOIs, or accession

numbers. For a list of recommended repositories, please see https://journals.plos.org/plosone/s/recommended-repositories.

If there are ethical or legal restrictions on sharing a

de-identified data set, please explain them in detail (e.g., data contain

potentially sensitive information, data are owned by a third-party

organization, etc.) and who has imposed them (e.g., an ethics committee).

Please also provide contact information for a data access committee, ethics

committee, or other institutional body to which data requests may be sent. If

data are owned by a third party, please indicate how others may request data

access.

Additional Editor Comments (if provided):

Reviewers' comments:

Reviewer's Responses to Questions

**Comments to the Author**

1. If the authors have adequately addressed your comments raised in a previous round of review and you feel that this manuscript is now acceptable for publication, you may indicate that here to bypass the “Comments to the Author” section, enter your conflict of interest statement in the “Confidential to Editor” section, and submit your "Accept" recommendation.

Reviewer #2: All comments have been addressed

Reviewer #3: All comments have been addressed

2. Is the manuscript technically sound, and do the data support the conclusions?

Reviewer #2: Yes

Reviewer #3: Yes

3. Has the statistical analysis been performed appropriately and rigorously? 

Reviewer #2: N/A

Reviewer #3: Yes

4. Have the authors made all data underlying the findings in their manuscript fully available?

Reviewer #2: Yes

Reviewer #3: Yes

5. Is the manuscript presented in an intelligible fashion and written in standard English?

Reviewer #2: Yes

Reviewer #3: Yes

6. Review Comments to the Author

Reviewer #2: The previous comments have been addressed. I have just a few additional comments:

1. The manuscript adheres to PRISMA-ScR reporting guidelines although on lines 79 and 80 of the revised manuscript, it states that ‘the methodology is based on Preferred Reporting Items for a Systematic Review and Meta-analysis of Diagnostic Test Accuracy Studies: The PRISMA-DTA’.

2. Table 1 is more of a critical appraisal than a quality assessment and the title should reflect that.

3. The legend states that the purple shape is 'unclear or moderate' while the JBI checklist rates the risk of bias as low, high, unclear or not applicable.

4. Lines 150 and 151 in Results section, 'hearing loss is first noted in adolescence...' - is this a finding or part of the discussion as the statement is referenced.

5. The layout (orientation) of Table 5 needs to change to Landscape as with Tables 4A & B.

Reviewer #3: Thank you for this important article highlighting a knowledge deficit in the actual incidence, prognosis, response to therapy of/for congenital syphilis. The recent increase incidence of congenital syphilis in the USA and Canada with unknown incidence in LMICs make filling this knowledge gap even more important as we move forward. I feel the reviewers comments/concerns have been adequately addressed. I do wonder why the authors chose not to have a second reviewer for articles in French. It would be informative to identify which articles were in French but not critical to do so.

7. PLOS authors have the option to publish the peer review history of their article (what does this mean?). If published, this will include your full peer review and any attached files.

Reviewer #2: **Yes: **Zainab Imam

Reviewer #3: No

---

## [Author Response · Author response to Decision Letter 1]

27 Feb 2024

Thanks for agreeing to review a revised version of our manuscript. We have addressed the suggestions as follows:

Please ensure that your manuscript meets PLOS ONE's style

requirements, including those for file naming. The PLOS ONE style templates can

be found at

and

Response: Changes were made to meet these requirements.

2. We note that your Data Availability Statement is

currently as follows: This is a systematic review so all data can be derived from the included manuscripts.

Please confirm at this time whether or not your submission

contains all raw data required to replicate the results of your study. Authors

must share the “minimal data set” for their submission. PLOS defines the

minimal data set to consist of the data required to replicate all study

findings reported in the article, as well as related metadata and methods

(https://journals.plos.org/plosone/s/data-availability#loc-minimal-data-set-definition).

Response: The manuscript contains all raw data required to replicate the results of the study. 

Response: Done

Reviewer #2: The previous comments have been addressed. I have just a few additional comments:

1. The manuscript adheres to PRISMA-ScR reporting guidelines although on lines 79 and 80 of the revised manuscript, it states that ‘the methodology is based on Preferred Reporting Items for a Systematic Review and Meta-analysis of Diagnostic Test Accuracy Studies: The PRISMA-DTA’.

Response: Thanks for noticing this error. The manuscript and the accompanying reference have been corrected. We realized that our abstract was not compliant with this reporting guideline so minor changes were made to make in the abstract it compliant. 

2. Table 1 is more of a critical appraisal than a quality assessment and the title should reflect that.

Response: The title was changed as suggested. 

3. The legend states that the purple shape is 'unclear or moderate' while the JBI checklist rates the risk of bias as low, high, unclear or not applicable.

Response: We omitted ”or moderate” which was an error.

4. Lines 150 and 151 in Results section, 'hearing loss is first noted in adolescence...' - is this a finding or part of the discussion as the statement is referenced.

Response: This is a summary of the results in the table. The sentence is now started with “In these reports…”. The reference was removed as it makes it seem like we are talking about a study that is not in the table. 

5. The layout (orientation) of Table 5 needs to change to Landscape as with Tables 4A & B.

Response: Done

Reviewer #3: Thank you for this important article highlighting a knowledge deficit in the actual incidence, prognosis, response to therapy of/for congenital syphilis. The recent increase incidence of congenital syphilis in the USA and Canada with unknown incidence in LMICs make filling this knowledge gap even more important as we move forward. I feel the reviewers comments/concerns have been adequately addressed. I do wonder why the authors chose not to have a second reviewer for articles in French. It would be informative to identify which articles were in French but not critical to do so.

Response: Only one of the authors is fluent in French. In the end, none of the studies in French met the inclusion criteria. Had there been many studies, we would have found another person who was fluent. 

Response: Done

---

## [Editor Report · Decision Letter 2]

21 Mar 2024

PONE-D-23-32597R2Scoping review of hearing loss attributed to congenital syphilis.PLOS ONE

Dear Dr. Robinson,

Thank you for submitting your manuscript to PLOS ONE. After careful consideration, we feel that it has merit but does not fully meet PLOS ONE’s publication criteria as it currently stands. Therefore, we invite you to submit a revised version of the manuscript that addresses the points raised during the review process.

 Please submit your revised manuscript by May 05 2024 11:59PM. If you will need more time than this to complete your revisions, please reply to this message or contact the journal office at plosone@plos.org. Please include the following items when submitting your revised manuscript:A rebuttal letter that responds to each point raised by the academic editor and reviewer(s). You should upload this letter as a separate file labeled 'Response to Reviewers'.A marked-up copy of your manuscript that highlights changes made to the original version. You should upload this as a separate file labeled 'Revised Manuscript with Track Changes'.An unmarked version of your revised paper without tracked changes. You should upload this as a separate file labeled 'Manuscript'.If applicable, we recommend that you deposit your laboratory protocols in protocols.io to enhance the reproducibility of your results. Protocols.io assigns your protocol its own identifier (DOI) so that it can be cited independently in the future. For instructions see: https://journals.plos.org/plosone/s/submission-guidelines#loc-laboratory-protocols. Additionally, PLOS ONE offers an option for publishing peer-reviewed Lab Protocol articles, which describe protocols hosted on protocols.io. Read more information on sharing protocols at https://plos.org/protocols?utm_medium=editorial-email&utm_source=authorletters&utm_campaign=protocols.

We look forward to receiving your revised manuscript.

Kind regards,

Bolajoko O. Olusanya, MBBS, FMCPaed, FRCPCH, PhD

Academic Editor

PLOS ONE

Journal Requirements:

Additional Editor Comments (if provided):

This manuscript is much improved now. However, substantial essential edits are required before it can be accepted for publication.

INTRODUCTION:

Paragraph 2; Lines 3-5: “In 2021 in Canada, the incidence of infectious syphilis in females was 729% higher than in 2017.“

- This statement is incorrect. Authors should check and revise the percentage increase from 7 in 2017 to 96 in 2021.

RESULTS

Lines 1&2: Authors please clarify: “Screening led to 154 records for full-text review of which 36 met inclusion criteria (Figure 1).“

- This statement is unclear. Authors need to describe the process of study were selection after removing duplicates to arrive at 154.

Screening of patients with hearing loss for congenital syphilis

Line 1: “There were 5 studies where patients with hearing loss for were screened for CS.”

- Delete “for” after hearing loss.

Table 1: Delete “were” from the subtitles under Type of study column e.g. Children with HL were screened for CS

Studies with interventions for hearing loss

Line 2: “All were observational and typically included a penicillin with addition of…”

- Authors to revise sentence

DISCUSSION

Opening sentence: “The scoping review shows that the literature on hearing loss due to CS is immature.”

- The word “immature“ in the opening sentence is unclear. Authors should rewrite and clarify.

Para 1, Last sentence: ”It seems unlikely that a systematic review would answers these questions”

- change “answers” to ‘answer’

Para 4, Lines 2 & 3: “It is possible that hearing loss started years prior but was not recognized.”

- Authors may also add the following sentence: “Particularly, if the hearing loss is delayed or progressive.”

GENERAL

The manuscript requires extensive language edits.
---

## [Author Response · Author response to Decision Letter 2]

22 Mar 2024

Response to reviewers:

INTRODUCTION:

Paragraph 2; Lines 3-5: “In 2021 in Canada, the incidence of infectious syphilis in females was 729% higher than in 2017.“

- This statement is incorrect. Authors should check and revise the percentage increase from 7 in 2017 to 96 in 2021.

Response: We decided that it was simpler and clearer to remove percentages and remove maternal Canadian data and present actual data for congenital syphilis for the same years in Canada and the United States. We therefore changed this section to read: “The number of cases of confirmed early CS born to women aged 15-39 years in Canada rose from 17 in 2018 to 117 in 2022 [4]. Trends in the United States (US) mirror this with an increase from 1325 congenital syphilis cases in 2018 to 3755 in 2022 [5].”

RESULTS

Lines 1&2: Authors please clarify: “Screening led to 154 records for full-text review of which 36 met inclusion criteria (Figure 1).“

- This statement is unclear. Authors need to describe the process of study were selection after removing duplicates to arrive at 154.

Response: Thanks for pointing out this error. It should have been 159 rather than 154 full articles that were pulled. We now added to the text “The figure outlines the reasons for exclusion of other records. “ We can outline the reasons in the text as well if the reviewer thinks that this is necessary.”

Screening of patients with hearing loss for congenital syphilis

Line 1: “There were 5 studies where patients with hearing loss for were screened for CS.”

- Delete “for” after hearing loss.

Response: Thanks for pointing out this error. It was corrected. 

Table 1: Delete “were” from the subtitles under Type of study column e.g. Children with HL were screened for CS

Response: Done

Studies with interventions for hearing loss

Line 2: “All were observational and typically included a penicillin with addition of…”

- Authors to revise sentence

Response: This was changed to “All were observational. Most commonly patients were prescribed penicillin with addition of prednisone followed by ACTH if response was poor or transient.”

DISCUSSION

Opening sentence: “The scoping review shows that the literature on hearing loss due to CS is immature.”

- The word “immature“ in the opening sentence is unclear. Authors should rewrite and clarify.

Response: We changed “immature” to “limited and low quality.”

Para 1, Last sentence: ”It seems unlikely that a systematic review would answers these questions”

- change “answers” to ‘answer’

Response: Done

Para 4, Lines 2 & 3: “It is possible that hearing loss started years prior but was not recognized.”

- Authors may also add the following sentence: “Particularly, if the hearing loss is delayed or progressive.”

Response: Thanks for the suggestion. We changed the sentence to read: “It is possible that hearing loss started years prior but was not recognized, particularly, if the hearing loss was slowly progressive.”

GENERAL

The manuscript requires extensive language edits.

Response: Many edits were made.

---

## [Editor Report · Decision Letter 3]

4 Apr 2024

Scoping review of hearing loss attributed to congenital syphilis.

PONE-D-23-32597R3

Dear Dr. Robinson,

We’re pleased to inform you that your manuscript has been judged scientifically suitable for publication and will be formally accepted for publication once it meets all outstanding technical requirements.

Kind regards,

Bolajoko O. Olusanya, MBBS, FMCPaed, FRCPCH, PhD

Academic Editor

PLOS ONE